# Social Media as a Tool for Informal Spanish Learning: A Phenomenological Study of Chinese Students’ Behaviour in Spain

**DOI:** 10.3390/bs14070584

**Published:** 2024-07-10

**Authors:** Xinyu Zhang, Francesca Romero-Forteza

**Affiliations:** Department of Applied Linguistics, Universitat Politècnica de València, 46022 València, Spain; fromero@upv.es

**Keywords:** informal language learning, social media, Chinese learners, Spanish language acquisition, cultural awareness, digital learning

## Abstract

This study explores the perceptions of Chinese learners in Spain regarding the use of social networks for informal Spanish language learning. The objective is to identify the challenges and benefits of using social networks to address the real needs of students in learning Spanish. A qualitative phenomenological approach was adopted, focusing on participants’ perceptions before and after using social networks. Eight Chinese students were selected for the study. The study was conducted in Valencia and Barcelona, Spain, from 1 September 2023 to 20 March 2024, and three commonly used social media networks were compared. The results indicate positive perceptions towards social media as a tool for learning Spanish, highlighting its usefulness in improving language skills and enhancing cultural awareness. Additionally, Xiaohongshu and Bilibili emerged as the most popular platforms for Spanish language learning among Chinese students. This study concludes that social media effectively meets the authentic needs of Chinese students learning Spanish in Spain, enhancing both language skills and cultural adaptation. This multifaceted approach reflects the complexity of learning Spanish in the digital age, combining personal passion, professional aspirations, and cultural adaptation needs.

## 1. Introduction

In recent years, as the exponential development of artificial intelligence in the 5G era has led to a faster pace of convergence in traditional media [1], mobile technologies for language learning have gained attention for their potential benefits for teachers and learners. Computer-assisted language learning (CALL) tools developed for mobile devices (MALL) have gained popularity due to their ability to create stress-free, self-paced learning environments, virtually unlimited input, and rich multimodal feedback [2]. Shuang [3] highlighted the importance of the Internet and online websites as inescapable elements of CALL in foreign language teaching and learning. The traditional teacher-centred model is inadequate for addressing the current challenges in foreign language learning due to its passive nature [4], limited interaction [5], lack of individualisation [6], insufficient opportunities to provide feedback [7] and resistance to change [8], thus necessitating a shift to more student-centred [9], interactive [10], and technology-integrated approaches [11,12,13] to improve language-learning outcomes. Nieto et al. [14] are of the opinion that determining how to develop learners’ autonomy in foreign language learning has been a major challenge, considering that teachers can use Internet resources to provide learners with rich and intuitive learning references for teaching, self-study, and improved teaching efficiency.

The global popularity of social networking sites has made them an important platform for information exchange and learning. For language learners in particular, social media not only offers the opportunity to interact with native speakers, but also opens up a whole new environment for informal learning. This paper focuses on a specific group (Chinese learners in Spain) and explores how this group uses social networking platforms for informal Spanish language learning. This group is uniquely characterised by the fact that these learners already have some basic knowledge of Spanish and most of them are studying at the master’s level in courses related to Spanish linguistics, which provides them with particular perspectives and experiences in using social media for language learning.

Through questionnaires and semi-structured interviews, this study aims to uncover the motivations, strategies, and effects of these learners’ use of social media as a language-learning tool. This research methodology allows for an in-depth understanding of learners’ individual experiences and perspectives, which provides insight into how social networks can be used effectively for language learning. Given the popularity of social networking in modern society, the findings of this study have important implications for language teachers, learners, and educational technology developers.

The core aim of this study is to explore the role and potential of social networks as informal learning environments for language acquisition while focusing on the limitations and challenges of this type of learning. By focusing on a group of Chinese learners in Spain, this study provides a unique perspective that contributes to a broader understanding of the use of social networks in intercultural language learning.

## 2. Language Learning through Social Media

In the last decade, digital applications have shifted from computer-based to mobile device-based formats, especially in the mobile Internet era [15]. In the field of language learning, mobile apps have replaced traditional websites as the preferred informal method for younger generations [16]. At the same time, mobile apps, represented by social networks such as Twitter, WhatsApp, and Instagram [17], play an increasingly important role, not only in changing the ways in which people communicate [17,18] and through which people can create personal profiles, post content, and facilitate the rapid exchange of information [17,19], but also in language learning, playing a key role in providing a platform for learners to share their experiences and communicate with one another [20].

With the popularity of smartphone apps, it has become a popular trend for language learners to share their learning experiences on social media [21]. New technologies and social media improve foreign language learning by providing access to information and enhancing motivation [22,23]. They improve global communication, interaction, and collaboration, offering advantages like idea sharing, time saving, and direct interaction [24,25]. According to Ramos [26], learners establish social and cultural links, sharing learning experiences and engaging in online interactions to acquire new information related to their learning, thereby motivating themselves and enjoying the process.

Behaviours [27] and decision-making [28] in language learning are also complex and include different stages (e.g., before learning, during learning) and types of decision-making (e.g., choice of learning resources, learning methods). When user-generated content is shared through social networking applications, it often stimulates people’s interest and motivation for language learning [29]. The advantage of social networks also lies in the frequency of user participation and interaction, as they attract everyday users to interact with fellow learners, online language teachers, and varied content, while also allowing users to create and share their own language-learning experiences [30].

However, learning a foreign language through social media is not without its obstacles, especially when it comes to achieving the specific learning outcomes expected. Although learners are exposed to more information about the language, teachers cannot take responsibility for the extent and reliability of the content posted. Ultimately, teachers may find themselves responding to questions posed by learners in any video or post that they see on social media, even if the learners are directed to what the teacher considers to be the most appropriate resource. It has therefore been argued that learning a foreign language through social media is only suitable for higher-level learners who are more capable of learning independently [31]. However, Arcy [32] has suggested various generational and age divisions, arguing that younger people tend to be more information and communications technology (ICT)-literate than older people, which suggests that, given that teachers are mostly older than their pupils, teachers have lower levels of ICT-literacy than their pupils.

Furthermore, Tao and Gao [33] noted that, during the coronavirus disease 2019 (COVID-19) pandemic in China, Internet users had searched for motivational content, including information related to language learning, due to the blockades and restrictions in China. Therefore, the impact of social networking applications and their role in stimulating language learning and influencing decision-making for different types of language learning deserve further investigation [34]. Weibo, Bilibili, and Xiaohongshu are currently the most popular social media platforms in China.

Weibo is a popular Chinese social networking platform similar to Twitter, allowing users to post short messages, photos, and videos. It is used for entertainment, sharing news, and educational purposes. The literature reveals that some Weibo systems, such as WeiboFinder, are effective for Chinese learners and outperform other methods in suggesting terms and documents related to the target word [35]; users can learn passively by following educational bloggers and browsing shared content [36]. Although there has been no direct research on the use of Weibo for foreign language learning, studies have shown the potential of Weibo in terms of information dissemination, social networking dynamics, and its role in public participation [37,38]; for example, the Instituto Cervantes in Beijing has an official account on Weibo, where it posts Spanish-related content. Weibo’s environment of interactions, topic tracking, and content recommendations supports language learning [39].

Bilibili is a Chinese video platform similar to YouTube, with 315 million monthly active users in 2023 [40], compared to YouTube’s 2.7 billion in 2024 [41]. Besides entertainment, Bilibili serves educational purposes, offering high-quality content, strong interactivity, and close-knit communities [42]. Wu [43] believes that cross-cultural communication through online media such as Bilibili can increase viewers’ knowledge and change their cognitive perceptions. There are a number of Hispanic bloggers, or Chinese bloggers residing in Spain, who share their life in Spain or their Spanish classes on this platform, and this intercultural learning activity enhances students’ Spanish as a foreign language (ELE) skill, intercultural communication skills, and knowledge sharing [44].Bilibili’s “Danmu” system, derived from the Japanese “danmaku,” allows real-time text comments to appear directly on video playback screens. This feature synchronises comments with video playback, enhancing interactivity. Viewers can post “Danmu” at any time, creating a unique social interaction that enriches the viewing experience, especially in anime, entertainment, and educational videos. The role of ”Danmu” systems in foreign language learning has been researched by scholars, with Bi [45] suggesting that ”Danmu” systems can lead to the learning having the illusion of a synchronous dialogue while watching videos, thus facilitating political and cultural communication. Mei [46] shows that “Danmu” makes a difference in user interaction both verbally and visually. All of the above researchers suggest that Bilibili holds potential for facilitating foreign language learning through interactive environments, intercultural communication, and rich language-learning resources.

Xiaohongshu is a leading Chinese social network and e-commerce platform focusing on user-generated content. Users share photos, videos, and text, and interact through likes, comments, and shares, creating a dynamic community. The platform values authenticity, encouraging real experiences and personal views. It collaborates with brands and influencers, attracting a diverse user base across ages and interests, including a large number of foreign users, making it an important platform for showcasing and sharing cross-cultural experiences. Although learning a language using Xiaohongshu has not been widely researched in academic works, some productive second-language learning methods are possible through Xiaohongshu. First, according to Boonma and Phaiboonnugulkij [47], instructional procedures based on the theory of multiple intelligences and incorporating learners’ individual characteristics can enhance the teaching and learning process in second-language classrooms. As a social networking platform, Xiaohongshu’s user-generated content and social interactions provide learners with authentic language environments and opportunities to practice communicative competence, such as by interacting, commenting, and posting. Secondly, a study by Feng et al. [48] shows that Chinese students learning Spanish adopt different learning styles and use all strategies with moderate or high frequency, with active style and metacognitive strategies influencing their academic performance. Xiaohongshu’s multimedia content, such as videos and images, can help learners better understand the application of language in different contexts and improve their memory and comprehension. In addition, following specialised language-learning accounts can provide access to useful materials, tips, and learning strategies. Finally, one researcher [49] suggests that the use of myths when teaching Spanish to Chinese speakers can overcome the difficulty of capturing their attention and introducing cultural content. The cultural and lifestyle content of Xiaohongshu can provide a similar cultural bridge. Immersion in the target language culture through Xiaohongshu content can contribute to a deeper understanding and learning of the target language.

There is a lack of specific research on the impact of Chinese social networks on Spanish language learning. Most studies have focused on the general impact of social networks on language learning, or on Chinese people learning other languages. Therefore, this paper aims to explore how Chinese social networks such as Weibo, Bilibili, and Xiaohongshu affect the learning of Spanish as a foreign language. This study will not only fill an important gap in the existing literature but will also provide a deeper understanding of the unique intersection of social networking technologies and language learning in the Chinese–Spanish context. By focusing on Chinese social networks and their impact on Spanish language learning, this study aims to reveal how Chinese cultural and linguistic characteristics affect the acquisition of a distant language such as Spanish. This approach could provide new insights into technology-assisted language learning, especially when cultural and linguistic differences play a key role.

## 3. Learning Languages in an Informal Context

Digital technology has expanded opportunities for second-language learning beyond the classroom [50]. Researchers and practitioners in ELE teaching and CALL are increasingly focusing on the informal digital learning of Spanish [51,52]. Colás-Bravo and Quintero-Rodríguez [53] define informal learning as personalised, self-regulated learning without institutional support, where learners decide what, when, where, and from whom to learn [54,55].

Previous research has examined the relationship between the time spent by learners of Spanish as a foreign language in informal learning activities and their outcomes in vocabulary, reading, listening, and speaking [56,57]. Recent studies [26,56,58] further highlight the diversity of informal foreign language learning and point out that these diverse activities have a significant positive impact on outcomes such as self-confidence, productive vocabulary knowledge, speaking, and improved performance on formal tests.

In more informal settings, Spanish language learning takes place through social networks rather than in traditional educational settings, which generates new considerations and possibilities. The growing demand for truly immersive foreign language learning, initially spurred by the global COVID-19 pandemic, has led to the sustained use of virtual platforms that capitalise on the dynamics of social networks for learning purposes. Although the acute phase of the pandemic has ended, the shift to online and hybrid learning environments has continued, reflecting a lasting transformation in educational practices [59]. Despite the obvious promise of social networks, there has been limited research on these networks in more casual contexts, and few studies have explored their structure, uses, and effective integration into the self-directed learning process, which is crucial for self-directed learning [53].

Given this absence, this paper aims to unveil the pedagogical relationships that can be established between social networks and informal Spanish language learning. The pedagogical structure of these networks ranges from linguistic content to interaction opportunities, providing an environment that can facilitate language learning in an authentic and flexible way. Exploring the potential of social networks for self-directed Spanish language learning in situations where the formality of the traditional classroom may not be the most appropriate option could open up new avenues of research and strategies for those seeking to master the language in a more flexible and adaptive way.

When the authors searched WOS, SCOPUS, and Google Scholar for social media (All Fields), Spanish learning (All Fields), China (All Fields), and student (All Fields), the search returned only one author within the last 5 years: Zhang Leticia-Tian, who conducted research on technology-assisted language learning (TALL) using Spanish-language social networking sites (RRSS) for Chinese students. Her study [58] highlights the potential of L2 (second-language) vlogs (video blogs) in language learning, especially the Spanish-themed learning community formed on the social platform Bilibili. Vlogs represent the complexity of learning styles, including attraction-oriented learning and self-controlled learning. Through daily video blogs, learners share their learning experiences, interact with their peers, are introduced to Spanish culture, and improve their vocabulary and oral expression [58].

## 4. Objectives

There is a growing body of literature on the implementation of TALL [60,61]; however, to date, there has been no study fully adapted to the context of Chinese students’ Spanish language learning in Spain to understand students’ perceptions of TALL through RRSS in an informal context. Despite the relevant academic background of the participants, this study makes a new contribution by exploring the challenges and benefits of implementation with RRSS. It focuses on settings where students face resource-poor constraints, such as limited access to authoritative resources, insufficient professional guidance, restricted interactive communication tools, and inadequate educational support for verifying the accuracy of information. These resource constraints are critical in understanding learners’ real needs in Spanish language teaching, particularly in the informal, self-directed learning environments that RRSS typically foster. Using a phenomenological research design, this study investigated participants’ perceptions before and after the use of RRSS sin order to explore how they perceived and implemented RRSS in terms of understanding students’ authentic needs. This study sought to explore the following research questions:(1)How do students learn Spanish through social networks?(2)What self-management/self-regulation strategies do students use to overcome challenges and frustrations?(3)What are students’ motivations for deciding to use social media to learn Spanish?

## 5. Methodology

### 5.1. Research Design

This study followed a qualitative phenomenological research approach. Phenomenological studies are studies revealing how people reflect upon their experiences, and how description, impact, and evaluation can be interpreted to determine how to approach a particular experience [62]. Data analysis for this research followed the guidelines and procedures set down by phenomenological studies on research questions, as communicated by Creswell and Poth [63]. This study seeks to investigate the perceptions and expectations, problems, and needs of Chinese Spanish learners using a phenomenological approach in a resource-poor context with the help of social media. A phenomenological approach was most appropriate for this study as it provided an opportunity to further explore the practices and needs of Chinese Spanish learners with social media. In addition, we consulted two experts in the field and asked them to use a Likert scale to rate the reliability of the survey questionnaire. The scale was as follows: Strongly agree (1 point), Agree (2 points), Neutral (3 points), Disagree (4 points), Strongly disagree (5 points). IBM SPSS v.27 statistical software (Armonk, NY, USA) was used to calculate Cronbach’s alpha and the Kappa index. The questionnaire had a Cronbach’s alpha of 0.784 and a Kappa value of 0.535, which describes the consistency of the ratings between the two experts. The inter-rater reliability reached a moderate level (for more details, please see the Table 1 below). This study was conducted from 1 September 2023 to 20 March 2024 due to the possible emergence of new social media platforms.

### 5.2. Participants

We selected eight Chinese students: four studying Internet and social networking in the MA course in Language and Technology at the Universitat Politècnica de València (UPV), and four studying social networking and media in the Spanish as L2 class of the MA in Spanish as a Foreign Language: Research and Professional Practice at the Universitat de Barcelona (UB). At UPV, there are 11 students enrolled in the course this year. However, this master’s program is globally oriented, so not all students are of Chinese nationality. Similarly, at UB, approximately 12 students enrol in the course each year, but only a few are Chinese. Chinese students applying for these master’s programs must have at least a DELE (*Diplomas de Español como Lengua Extranjera*) C1 or B2 Spanish proficiency and an interest and academic background in social network language learning. Therefore, we could only select Chinese students who met these stringent criteria. Although the sample size is limited, these eight students all meet the rigorous selection criteria, ensuring that their academic backgrounds and research interests are highly relevant to the study. We surveyed these diverse participants to understand students’ learning experiences and needs in resource-poor situations. To select the respondents, all had to be graduates in Spanish or Hispanic Philology from a Chinese university. After collecting verbal consent from all participants, there were six women and two men, and their average age was 24 years (with an age range spanning from 21 to 26). For the master’s degree program in Language and Technology at the Universitat Politècnica de València (UPV), there was one woman who was 41 years old; this is of note because age is an influential element of digital competence in teaching [64].

An interview outline (Appendix A) was sent to the participants via WeChat to engage in semi-structured interviews. Forms written in both Spanish and Chinese were sent to the participants to give them a preview of the questions and, because the participants were native Chinese speakers, having the Chinese translations attached to the forms helped the participants to better understand the questions and to provide detailed answers. Semi-structured interviews were most appropriate for this study because their design allowed both the interviewer and interviewee to expand on the questions and answers more in depth. This also helped to guide the participant regarding what to consider in the two-way interaction format [65], which is missing in a one-way directional approach, i.e., reflections. As such, this method allowed the author to probe further into the questions of why, what, how, or when. When we initially contacted the interviewees, we informed them that we would be conducting anonymous interviews, and each interviewer signed an informed consent form (Appendix B).

## 6. Results and Discussion

### 6.1. Students Learn Spanish through Social Media and Assess the Effectiveness of Learning Spanish through Social Media

On a scale of 0–10, how would you rate the effectiveness of learning Spanish through social media? Reasons? (The results can be seen in the Table 2 below).

#### 6.1.1. Why Do You Want to Learn Spanish? Do You Want to Work as an ELE?

Two of our eight interviewees are interested in Spanish as a hobby and would like to live in Spain in the future, and one of them mentioned that he was particularly interested in Spain because of its victory in the 2010 World Cup. The other five think that knowing Spanish would give them more job opportunities, and all gave the example of becoming an ELE teacher to further promote Spanish language and culture in China, where Spanish is still a minority language that few people know. Finally, there was a special case of an interviewee who studied a degree in Spanish involuntarily due to requesting the wrong order of study preference upon his enrolment to the Chinese university, but after studying it he realised the importance of promoting Spanish language and culture in China, and this is why he wants to promote Spanish language and culture through the use of technology.

#### 6.1.2. Through Which Social Networks, Both Chinese and International, Have You Learned Spanish? WeChat, Xiaohongshu, Bilibili, TikTok, Instagram, Facebook, etc. The Differences between Learning on Chinese and Non-Chinese Social Networks

WeChat is a widely used social network in China. All of our respondents use WeChat as a social media platform, and two of the interviewees are using WeChat to learn Spanish, where they joined a Spanish language-learning group that usually has an administrator and a teacher who supervise the students in the group learning Spanish. This group has an average of ~300 students, but according to the descriptions of these two respondents, its learning environment is not active. Of the eight students interviewed, all had experience of using Xiaohongshu to learn Spanish. Participant 4M UPV said that “I personally use Xiaohongshu more because my aim is to pass the exam and there will be more resources and experiences in Xiaohongshu”. Participant 6M UB and Participant 8F UB felt that there was a lot of Spanish content on Xiaohongshu, such as culture, food, travel, etc., and that Xiaohongshu was a better social network to study and learn Spanish language and culture.

Participant 7F UB said, “*I think I am better at using Xiaohongshu to learn Spanish because there are many Spaniards and Latinos on the platform, and they share videos and some textual content on Xiaohongshu, which are usually subtitled in Chinese, which is easy for me to understand*”.

Four students said that they would use Bilibili to learn Spanish because Bilibili allows them to comment on the videos in real time: “I can see comments from other users who are watching the videos with me” (Participant 6M UB and Participant 3F UPV). All of the interviewees said that they think that Bilibili has a competitive advantage, as all of the videos are long enough that they feel like recorded lectures, they can understand the information in more detail through the long videos, and there are more examples presented in the videos than those on TikTok, Instagram, and Weibo.

Three students had used TikTok to learn Spanish, but one of them ended up quitting and using it only as an entertainment social network because he thought that there were too many entertainment elements on TikTok, and it was too easy to become distracted (8F UB). Another student thought that TikTok was too commercialised, with bloggers using short videos to attract traffic with the ultimate goal of selling their Spanish lessons, which the student thought was not in line with their original intention, so they also gave up using TikTok to learn Spanish (2F UPV).

As for other non-Chinese social media, all of the respondents said that they typically only use Instagram and, in addition, they like to see the content posted by Spanish influencers on Instagram to learn Spanish indirectly (8F UB).

#### 6.1.3. What Are Your Expectations If Teachers Use Social Media to Teach in a Master’s Classroom? What Successes and Challenges Do You Think You Will Encounter When Using Social Media to Learn Spanish?

All of the students said that they would like their teachers to share some of the Spanish teaching bloggers on social media and teach them to recognise which bloggers are sharing quality content, because it is difficult to know who is sharing more quality content.

*One of the recurring expectations from the initial survey was that social media would provide opportunities to “interact with native Spanish speakers”. I think it can facilitate our interactions with other students, but my problem is that I get distracted particularly easily, so I would like my teacher to allow my classmates to participate when using social media to teach classes*.(1F UPV)

They also expressed a desire to use social networks to teach Spanish language and culture, making learning more fun with graphics, videos, and comments. Participant 2F UPV said, “Self-monitoring and self-regulation of students’ learning through social media is also a very important challenge”.

#### 6.1.4. In the Context of COVID-19, Is There an Impact of the Use of Social Media for Learning Spanish, and Do You Think It Is Significant?

After talking to several students, it was found that COVID-19 has had a significant impact on promoting the use of social media to teach Spanish. Online platforms and social media have become important tools for teaching and learning languages, due to limitations on mobility and face-to-face teaching. The need for virtual connectivity has prompted teachers and learners to explore and adopt digital methods. In Spanish language teaching, social networks offer a practical and communicative platform for language practice. Online interaction, whether through posts, comments, or chats, allows students to improve their language skills in an authentic context. In addition, authentic Spanish content such as news, blogs, and videos shared on social media provides learners with greater exposure to the language and culture. The pandemic has led to a significant shift in the use of social media in Spanish language teaching, making it a valuable learning resource in the digital environment. However, three of our respondents expressed the pessimistic view that, for the moment, social networks can only be used for informal learning, and that although COVID-19 has accelerated their development in language teaching, there is still a need to regulate what is learned, how it is taught, the quality of teaching, etc.

### 6.2. Students Utilise Social Media for Self-Management and Self-Regulation Strategies and Evaluate the Effectiveness of Social Media in Helping Overcome Challenges and Frustrations

On a scale of 0–10, can the use of social media help students with self-management/self-regulation strategies to overcome challenges and frustrations? Reasons? (The results can be seen in the Table 3 below).

#### 6.2.1. What Characteristics Help Learners to Self-Monitor/Self-Regulate through Social Networks?

Four students (3F UPV, 5F UB, 6M UB, and 8F UB) indicated that they use the apps’ content filtering and content management features, and that with the content filtering feature they can choose to unfollow or block certain types of content, which helps to minimise distractions and avoid unwanted influences. In addition, 1F and 2F mentioned that they use time tracking and reminders, and that some social networks offer time tracking features that allow users to see how much time they spend on applications. Usage time reminders can also be set to help students be aware of and monitor the time that they spend on social networks. Participant 6M mentioned that he followed some learning bloggers and only browsed content posted by these bloggers, that some of them opened study groups on social media platforms, and that this kind of community or group could help students to stay motivated and focused.

There were also two students (4M UPV and 7F UB) who said that they try to be Spanish bloggers by sharing the Spanish knowledge that they consider important and, at the same time, they participate in this as a good form of self-monitoring because they think that when users read their posts they will give them feedback (likes, favourites) or comments, and they will be able to judge the quality of the posts that they publish by the number of likes and favourites, while simultaneously exchanging their knowledge with others in the comments, and some of them also share their learning experiences in the comments. When the author asked the other six students if they had ever shared their learning experiences on social media, five of them had had the idea but had not yet started to put it into practice; some thought that their Spanish was not good enough, while others felt that they lacked motivation to do so, but most of them said that they would try to post on social media in the future.

#### 6.2.2. Can You Compare and Contrast the Contribution to Spanish Language Learning before and after the Use of Social Media?

Many students agree that social networks offer an enriching perspective on learning Spanish, improving vocabulary, writing, and listening comprehension, fostering autonomy and access to materials, enhancing social and cultural skills, or even interacting with native speakers and language culture, as opposed to the traditional classroom.

*Platforms like Xiaohongshu can improve writing skills and vocabulary, when I went to university in China, due to Chinese internet restrictions, there were some Spanish newspapers (EL PAIS, EL MUNDO, etc.) that we could not browse, so, through the transport of some bloggers, we could read these authentic materials. However, due to our limited level of Spanish, even though we are now studying in Spain without internet limitations, we still cannot understand the meaning of many phrases or words by reading the real material, but through the annotations of these bloggers we can read these news and understand them well, which I think is more effective than if I had gone to the news sites to find the information by myself*.(6M UB)

*When I was studying in China, there was a course called Reading Spanish Newspapers, in which the teacher would download some editorials from Spanish news websites for us to read in class, and the teacher would explain them afterwards. But now, with social media, I can learn the points of these newspapers by myself, and at the same time I can discuss them with other users or bloggers in the comment section, which I think has the same effect as the undergraduate course I took then, but with more freedom of time, and more autonomy and flexibility in choosing the reading materials*.(8F UB)

*It has developed my ability to learn on my own, I can access material on social networks at my own pace, for example, Bilibili offers 0.5× speed, 1.25× speed or 2× speed, when watching videos I can choose slow or fast depending on my level of understanding, although I spend 30 minutes a day watching or listening to videos, but the benefits are great and I am gradually accepting the speed of Spanish speech in real life*.(9F UB)

*Although we live in Spain, we don’t have many ways to know about local festivals due to information barriers, but some bloggers who live in Spain can help us to remove these barriers, like Valencia Secreto on Instagram, or some Chinese bloggers in Xiaohongshu who share some festivals in their local area, I also went to Buñol town to attend the tomatina festival after reading their posts. I was very impressed with this festival, I had learned about it before in China, but I was very impressed with this festival, I knew about it before in China, but I couldn’t attend live, and I am glad I didn’t miss the festivities after reading these bloggers’ posts*.(4M UPV)

#### 6.2.3. What Measures Have Teachers Taken to Ensure the Effectiveness of the Language Forms and Functions, Content Knowledge, and Skills You Have Learned? Can You Give Some Examples? What Improvements Still Need to Be Made?

Teachers start by clarifying the objectives of the class and complement their classroom content with social media as a way of facilitating the connection between the course content and the technology platforms that students use on a daily basis. This not only gives students a sense of integration but also ensures that what they learn is not isolated from the real world. For example, teachers can discuss a particular language point in class and then ask students to find relevant topics to discuss on social platforms, so that they can apply what they have learned in real contexts. They can also encourage students to use tools such as WeChat group chats in their classwork to support independent learning and track their progress. Students will also return to class to confirm with the teacher the accuracy of what they have learned after applying their classwork on social media, which further deepens their learning.

However, there is still room for improvement. For example, the relevance of course content to students’ personal interests could be improved in order to increase motivation to learn. In addition, the articulation of learning activities outside the classroom with classroom learning could be strengthened to ensure that students are able to take the knowledge and skills learned in the classroom and apply them more effectively to their interactions in social networks and real-life situations. Finally, teachers can further explore how to use technological tools to support students’ distance and self-directed learning, particularly to provide personalised learning resources and feedback.

### 6.3. Students’ Motivations to Use Social Networks to Learn Spanish

In the case of participant 1F UPV, who is very active in learning about the use of social networks since she devotes considerable time to social media usage, she searches for people in a similar situation to her own through social networks and learns by understanding their learning patterns.

All interviewees mentioned that there are many free resources that can be found through social networks (novels in Spanish, Spanish manuals, films in Spanish, etc.), but their biggest concern is the issue of copyright, which is a major challenge for education in social networks.

When analysing the various influences of social networks on the foreign language-learning habits of the Chinese, we can take into account specific cultural characteristics, such as the Chinese preference for the use of idioms in their communication. However, it is difficult to find exact Spanish equivalents for many expressions (or idioms). Sharing on social media can bridge this gap, as users share content that helps learners understand how to convey similar meanings in English, thereby making their expressions more authentic.

Moreover, there is a lack of a linguistic environment for Chinese students—even those who come to Spain often shy away from communicating with locals for fear of their accent and pronunciation. To overcome this obstacle, they hope to correct their pronunciation by imitating the pronunciation of characters in social media videos. This imitation exercise not only helps to improve the accuracy of their pronunciation but also increases their confidence in the language.

In their learning practice, some students have found that learning a foreign language through social media not only increases learning flexibility but also improves the efficiency at which language-learning resources can be found. This is due to the advanced search capabilities of social media platforms, which use algorithms to quickly locate and filter specific information that learners are interested in or need, in contrast to the step-by-step learning model of the traditional classroom environment. Students can choose their own content based on their learning pace and points of interest, making the learning process more personalised and efficient.

On the other hand, one student also mentioned an innovative feature recently introduced by the video-sharing platform Bilibili: the use of artificial intelligence technology to summarise video content. The application of this technology has significantly reduced the time spent selecting educational videos. Students can quickly determine the relevance and value of videos based on the content summaries provided by the artificial intelligence (AI) and then choose to view those videos that best meet their learning needs. This approach not only improves the relevance of learning but also avoids the time-consuming task of blindly searching through a large amount of content.

## 7. Conclusions

### 7.1. Social Networking Preferences

In investigating Chinese students’ Spanish language learning, we observed that they employed a variety of strategies (for more details, please see the Table 4 below), including social media platforms. These strategies not only reflected their cultural and educational backgrounds but also demonstrated their unique ways of adapting to and using different social media features. The students used a variety of Chinese and foreign social media platforms, each of which offered different benefits. Gao and Feng [66] found that different social media platforms offered varied benefits, with users of social networking sites motivated by social interaction, whereas users of microblogs were motivated by self-expression and information-seeking. In the case of the eight students surveyed here, they all indicated that the social networking applications that they use most to learn Spanish are Xiaohongshu and Bilibili, and that they do not use TikTok because it is too commercial and entertaining, making it difficult for them to focus on learning the foreign language. It is worth noting that Chinese learners tend to use Chinese social networks even when they are abroad, probably because they are more familiar with Chinese interfaces and content, such as Chinese subtitles or Chinese annotations. On platforms such as Xiaohongshu, students focus on posts containing language information and follow bloggers who specialise in posting foreign language-learning content. This suggests that learners are more likely to browse educational content and personal brands on Xiaohongshu [67]. In contrast, students engage more in social interactions and comments on the Bilibili platform, which offers entertainment content such as Spanish-language movie clips, songs, and celebrity interviews. This preference for different social platforms and the variety of content they provide (instructional or entertainment) will have a deep-seated impact on users’ perceived usability of every such platform.

### 7.2. Evaluation of Effectiveness

The students’ perceptions of the efficacy of learning Spanish through social networks was also combined, resulting in both favourable and moderately favourable scores, with the average score varying from 4 to 9 out of 10. The students noted the advantages, among others, were due to access to more resources, including cultural experiences, varied vocabulary, and authentic materials. The learners were of the view that the social networks were useful tools to develop cultural awareness and the use of the target language as they began to take courses in Spanish [73]. Access to these platforms is of particular use to them if they wish to undergo cultural immersion or prepare for exams such as the DELE or SIELE. The use of authentic content, including the use of colloquialisms and slang, is of particular value in terms of authenticity and practicality of the context of use. Siddig [74] pointed out that social networking facilitates interactions with native speakers and immersion in the culture, improving diction and pronunciation. However, challenges have also been identified, such as the lack of a formal teaching structure and the distractions inherent in social platforms. In academia, there are concerns about the quality of the content on social media and the ability of students to self-regulate their learning. Thus, during the pandemic, China’s political isolation accelerated the adoption of social media in Spanish language teaching, offering a practical alternative for linguistic interaction at a time when face-to-face education was limited. As a result, students used social networks to connect with the university community [75], to collaborate in learning, and to engage in interactions with peers and teachers [76].

### 7.3. Self-Monitoring/Self-Regulation Strategies

The students’ perceptions of the role of social networks in self-monitoring varied widely, ranging from 3 to 9 out of 10. This variation suggests that social networks offer benefits as well as significant challenges.

In exploring students’ Spanish language learning through social media, particular attention is paid to how students use self-regulated learning strategies to improve their concentration and time management skills. As Zhou et al. [77] point out, there are four common self-regulated learning strategies that students apply in the learning process: they set clear learning goals, create a learning environment that is favourable for study, monitor the learning process, and conduct self-assessments. All of these strategies combine to make it possible for students to study effectively in social networking environments in which they have to multi-task or be distracted. Regarding practical concerns, students might have certain tools for time management, such as alarm clocks or digital timers, to help them plan the time they devote to social media. This strategy works to avoid over-browsing and information overload.

Apart from time management, Zimmerman and Martinez-Pons [78] pointed out several important characteristics of self-regulated learners, such as high self-efficacy, independence, and intrinsic motivation. Such characteristics can enable students to plan and organise their learning process more effectively and to better regulate and evaluate themselves. In addition, learners filter social media content by creating collections and bookmarked lists. They realise the importance of screening and selecting the content relevant to their learning goals to avoid distractions and improve learning results. This can not only help students to concentrate, but also assist in the orderly processing of information and effective review. The combination of audio–visual elements, such as video and audio materials, has also been proven to improve students’ concentration in the learning process. Abundant audio–visual content on social media channels can trigger students’ interest as well as provide a more intuitive and interactive learning experience [79]. For example, watching original Spanish videos or playing interactive language-learning games can strengthen learners’ motivation and participation [80].

Research by Sanz de Acedo Lizarraga et al. [81] also demonstrates that learning the skill of self-regulation not only enriches academic performance, but also develops self-reflectivity, self-investigation, self-assurance, and empathy to a substantial extent among those students at the secondary level who have social maladjustment to cope with. This shows that the pedagogical significance of self-regulated learning strategies is well beyond the academic arena and is closely connected with overall learner development.

Finally, to achieve the meaningful use of social media for education, one may also need to improve the relevance of curriculum content, establish links between in-class outside-class learning, and further explore how to learn using technology with regard to student self-directed learning. This can include sorting out the curriculum content that is closely related to each student’s personal interests and career. This also aids in the development of tools and technical platforms that help students learn independently outside the classroom. These developments not only improve the relevance and attraction of learning, but also cultivates self-exploration and critical thinking skills in students.

### 7.4. Motivations for Learning

The motivations for learning Spanish using social networks are diverse and strong. One of the most important motivations is a personal interest in the language and culture of Spanish-speaking countries, which reflects an interest in and appreciation for the culture and diversity of the Spanish-speaking world. Learners are positive about the usefulness of learning Spanish, and they act, especially to compensate for the lack of time, to develop their language skills, and knowledge of Spanish-speaking countries and current affairs, all of which are valuable tools for raising cultural awareness and targeting language use [73]. Performance expectations, effort expectations, social impact, facilitative conditions, enjoyment, and habits are other reasons students use social networks [82]. Another motivator is the opportunity to work teaching Spanish as a foreign language, which many learners strongly consider. Consequently, the motivation to learn Spanish stems from the importance of the Spanish language in the actual global context and the necessity of languages in professional life, among others.

Cultural integration and adaptation to life in Spanish-speaking countries are also prime motivators. For students living or planning to live in these countries, learning Spanish is not only an academic activity, but a practical necessity that facilitates daily communication, cultural understanding, and integration into society. Combined, these motivations paint a picture of learners motivated by a mix of personal passion, professional aspirations, and cultural adaptation needs [83], demonstrating the multifaceted nature of Spanish language learning in the digital age.

In addition, social networks offer greater flexibility and efficiency in learning, allowing learners to select specific content that aligns with their interests and pace of learning, making the process more personalised and efficient. Innovation and technology also play a key role; for example, platforms such as Bilibili use artificial intelligence to summarise video content, making it easier for learners to quickly select relevant and quality materials, and reducing the time spent searching through large amounts of content [84].

Although there are currently few studies on the use of social networks by Chinese students for learning Spanish, Ahmad et al. [85] examined the role of social media in Spanish language learning in the Indian context. Their study found mixed evaluations of social media’s effectiveness in language learning, with different platforms offering distinct advantages and challenges for achieving various learning goals. Similarly, our study found that Chinese students face challenges but also benefit from the diverse resources and interaction opportunities provided by social media in resource-limited environments. Comparing these studies helps us to understand the impact of social media on Spanish language learning across different cultural contexts.

## 8. Limitations and Prospects

This paper explores the use of social networks for informal Spanish language learning among Chinese students in Spain, but it has several important limitations. First, the sample size was only eight students, which limits the generalisability and representativeness of the results. These students already had some intermediate knowledge of Spanish and may not be representative of students at all language levels. Second, as the research methodology consisted of questionnaires and semi-structured interviews, the data may have been affected by subjectivity and self-report bias. In addition, the diversity of social media platforms and usage habits, along with their changes over time, may limit the results of this study to a specific time and context.

Future research could extend the findings of this study in several ways. First, increasing the sample size and including students with different levels of proficiency in Spanish could increase the generalisability and depth of the results. Second, exploring the use of different social networking platforms and how they affect language learning would contribute to a more comprehensive understanding of informal learning environments. In addition, the use of different research methods, such as observational or experimental methods, could provide more objective data on learning effects and processes.

## Figures and Tables

**Table 1 behavsci-14-00584-t001:** Inter-Rater Reliability Analysis Results.

Question Number	Expert A	Expert B
1.	2	2
1.1.	3	3
1.2.	4	4
1.3	3	3
1.4.	2	2
2.	2	3
2.1.	2	2
2.2.	3	3
2.3.	3	5
4.	2	3

**Table 2 behavsci-14-00584-t002:** Participants’ Ratings and Reasons for Learning Spanish through Social Media.

Participant	Score	Reasons
Participant 1F UPV	7	*“* *From my experience, I learn Spanish through social media for DELE (Diplomas de Español como Lengua Extranjera)/SIELE (Servicio Internacional de Evaluación de la Lengua Española) exam preparation, students share their exam experiences and some questions asked for me are very useful”.*
Participant 2F UPV	8	*“* *I like to use social media to learn vocabulary, because I always struggle to remember words due to being older, so I follow a few bloggers who share words on social media and check them out when I have time, mostly browsing short videos and images.* *I think it’s very effective because I’m not very good at studying on my own and finding materials, so I like to follow a few quality bloggers and learn from the valuable content they share”.*
Participant 3F UPV	4	*“* *Although I use social media to learn Spanish, learning Spanish on social media can lack systematic formal instruction for me compared to teacher-led lessons, which can be challenging for me. In addition, I often get distracted because the system’s algorithm offers me not only language learning content, but also some other content that interests me, so I quickly forget about it after reading it”.*
Participant 4M UPV	6	*“* *There are many learning resources on social media, and I usually bookmark them in my account, but over time, I have more and more posts bookmarked in my account, and I find that I don’t read them all, and I can’t remember what I’ve bookmarked, and I’m disgusted by my favourites and don’t want to open them. I only remember words or phrases made up“.*
Participant 5F UB	9	*“* *This method of learning is still useful. When I take the metro, I watch some short videos in Spanish, mainly about Spanish culture and Spanish social issues. Because the videos have both Chinese and Spanish subtitles, and if I don’t understand them, the Chinese subtitles can help me understand better. In fact, when I tell my Spanish friends about the cultural stories in the videos, they all tell me that it is true and I feel happy because it is a way to integrate into the local culture”.*
Participant 6M UB	8	*“* *As I am in Spain, I can read the comments left by others on Instagram, many of these comments are informal colloquialisms or social media slang, when I see them I guess the meaning of the words in the context of the video and google their explanations and confirm them with my Spanish classmates, after which I will use them in my daily life, which I find very interesting”.*
Participant 7F UB	8	*“* *I would use Chinese social media apps to learn Spanish because they usually have translated subtitles and I can better understand what they are saying. The bloggers I follow basically have DELE C1-C2 language certificates, so I trust the accuracy of what they share, and I also live in Spain now, so I have some discernment. As I believe that it is very worthwhile to learn from what others share, I always find new learning resources, such as websites for learning, vocabulary and Spanish American culture”.*
Participant 8F UB	7	*“* *When I first started learning Spanish, I didn’t like using social media because I didn’t understand the content, I didn’t find it interesting and it was bad for my self-esteem. However, as my Spanish improved, I realised that I was able to understand the content that was shared on social media, so I think it helped me to increase my interest in learning and boost my self-esteem”.*

**Table 3 behavsci-14-00584-t003:** Participants’ Ratings and Reasons for Using Social Media for Self-Management and Self-Regulation.

Participant	Score	Reasons
Participant 1F UPV	8	*“* *I think yes, many students share on their social media accounts how they plan to prepare for exams and even share what apps they use to stay focused on their studies, so learning about other people’s experiences through social media can be very useful. So my strategy is to use some focus tools”.*
Participant 2F UPV	3	*“* *This is not a problem for me because I limit myself to spending a maximum of 30 min on social media, and memorising words requires extreme concentration on my part, so I don’t worry about getting distracted”.*
Participant 3F UPV	9	*“* *In fact, the system’s algorithm throws me into a lot of content that is not related to language learning and I can’t focus on a single entry. So what I do is choose some videos and use a combination of audio and visual to help me concentrate”.*
Participant 4M UPV	4	*“* *Distraction is not a problem, my biggest problem is the inability to keep learning fresh. On social media, I can’t review what I learn on a regular basis, I have to deliberately search for what I have seen before, and I don’t think there is anything planned or logical about this kind of informal learning, so I have created a list of favourites, and I have put useful content in my favourites, so that it is easy for me to review it”.*
Participant 5F UB	4	*“* *The content I look at is not particularly intellectual in itself, so it’s easy for me, and I’m interested in the cultural side of things. If I were to read something grammatically intellectual on a social network, I’m sure I’d be distracted”.*
Participant 6M UB	6	*“* *I like to immerse myself in foreign languages to avoid distractions so I don’t like using Chinese apps, but recently I’ve noticed the same problem on Instagram, the algorithms seem to automatically recognise that I’m Chinese and start throwing Chinese published content at me, but that’s not what I want and I don’t know how to avoid it. What I have done is to follow a few Spanish learning bloggers who are Spanish or South American so I can go to their page and focus on their content”.*
Participant 7F UB	9	*“* *It’s a big problem for me, I can’t get rid of the distraction of other social media posts. But I don’t really mind, because going through social media is a relaxing thing for me, and I’m pleased to be able to learn some Spanish in this informal learning context”.*
Participant 8F UB	8	*“* *There will be distractions, that’s inevitable. But when I read something interesting, such as a cultural introduction, a story about a famous person in Spain, an allusion to some Spanish slang, and a commentary on a social event in the Spanish media, etc., I will be focused”.*

**Table 4 behavsci-14-00584-t004:** Strategies Employed by Chinese Students in Learning Spanish.

Strategies	Explanation
Memorisation Techniques	Chinese students often employ memory strategies such as repetition of words, use of flash cards, and note memorisation to enhance vocabulary and memory of new language elements [68].
Cognitive Strategies	This includes generalising and summarising what has been learnt, inferring the meaning of new words through context or known knowledge of the language, and applying and reinforcing the language concepts learnt through practical engagement in problem solving [69].
Metacognitive Strategies	Learners devise their approach to learning, establish measurable learning objectives, self-evaluate their learning process, measure and assess their own performance, and make adjustments to their learning strategies as necessary [70].
Social Strategies	These can include seizing the chance to converse with peers, educators, or with actual Spanish speakers and immersing themselves in a language exchange endeavour so that their listening and speaking skills can be improved [40].
Affective Strategies	Students apply various tools manage their emotional state and motivation, including reward systems, relaxation strategies targeting the reduction of anxiety, and a positive attitude towards learning Spanish [71].
Compensation Strategies	To compensate for students’ deficiencies in the mastery of the language, they speak in Spanish through the use of gestures, context clues, and making educated guesses to attempt to communicate and understand the language being spoken [72].

## Data Availability

Data are contained within the article.

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
