# Peer review of "Social Media as a Tool for Informal Spanish Learning: A Phenomenological Study of Chinese Students’ Behaviour in Spain"

_behavsci, 2024, doi:10.3390/bs14070584_

Round 1
Reviewer 1 Report
Comments and Suggestions for Authors
The paper investigates the impact of social media on the learning behaviors of Chinese individuals studying Spanish, focusing on three main questions. 1. How do students learn Spanish through social networks? 2. What self-management/self-regulation strategies do students use to overcome challenges and frustrations? 3. What are students’ motivations for deciding to use social media to learn Spanish? The authors adopt a qualitative phenomenological research approach.
This investigation is interesting and would have impact on behavioral studies in learning foreign language.
However, I have the following suggestion.
1. Cronbach's alpha and the kappa index should be provided in details.
2. Section 2 " Language Learning through Social Media" and 3 "Learning Languages in an Informal Context" are too verbose. I suggest the authors condensed these two sections.
3. The organization could be improved by moving 7.5 Limitations and prospects as a separate section.
Comments on the Quality of English LanguageL696 and 701, improper symbols appear. Similar situations should be addressed in the following lines.
A few sentences are too long, such as L267 - L272.
Author Response
Point-by-point response to Comments and Suggestions for Authors
Comments 1: Cronbach's alpha and the kappa index should be provided in details.
Response 1: Thank you for pointing this out. We agree with this comment. Therefore, we have refined section 5.1 of the article, adding details on Cronbach's alpha and the kappa index, and attached the scoring results of the Likert scale to the article. You can find these changes on page 6, lines 259-267. The scoring results of the Likert scale can be found on page 6, lines 269-270.
Comments 2: Section 2 " Language Learning through Social Media" and 3 "Learning Languages in an Informal Context" are too verbose. I suggest the authors condensed these two sections.
Response 2: Agree. We have, accordingly, condensed sections 2 "Language Learning through Social Media" and 3 "Learning Languages in an Informal Context." Each section has been reduced by one-third of its original content to enhance clarity and conciseness.
The specific changes are as follows:
Section 2: The revised content can be found in lines 69-150.
Section 3: The revised content can be found in lines 186-191.
Comments 3: The organization could be improved by moving 7.5 Limitations and prospects as a separate section.
Response 3: Agree. We have made the necessary adjustments. This section has now been revised as section 8. The specific change can be found on page 16, line 647.
Response to Comments on the Quality of English Language
Point 1: L696 and 701, improper symbols appear. Similar situations should be addressed in the following lines.
Response 1: Thank you for your feedback regarding improper symbols. The lines you mentioned (L696 and L701) are now L685 and L690, respectively. Our questionnaire is composed in English, Spanish, and Chinese. The parts you pointed out are in Spanish, and we have reviewed these sections for improper symbols.
Spanish orthography includes special characters such as inverted question marks (¿) and accent marks (e.g., qué, através). These are integral to the language and are not improper symbols. After careful review, we did not find any improper symbols in the specified lines.
If you could provide more specific details about the issues you observed, we would be more than happy to correct them. We appreciate your attention to detail and will continue to ensure the accuracy of our manuscript.
Point 2: A few sentences are too long, such as L267 - L272.
Response 2: We have addressed this by splitting it into two sentences. The revised content can be found in lines L232-237.
Reviewer 2 Report
Comments and Suggestions for Authors
The article fills a gap in the literature, by analyzing the role of social media in language learning environments from the point of view of the users. Although this topic has strongly been explorated in recent years, due to the spread of online platform and their use in the LS classroom, the authors pointed out a lack of studies specifically dedicated to Chinese learners of Spanish, and on the role of social media use in helping students engaging in learning activities.
The article is very well structured, and the literature review covers all the possibile topics lately added in the final discussion. Methods and materials are carefully explained, and the different Appendix are incredibly useful to further scholars approaching similar topics maybe in other contexts.
I found particularly interesting, although maybe a bit too long in the general economy of the paper, the presentation of the different social networks available to the students and tested in this paper.
Generally speaking, therefore, the paper represents a huge contribution to the scholarship. However, I believe that the final version could benefit of some small changes as in the following suggestions.
Suggestions
1) The corpus is indeed quite limited since 8 students couldn't possibly be representative of the whole category of Chinese learners of Spanish. The author explicitely address this poin in section 7.5, but I suggest to already touch this point when presenting the dataset, by also clarifying why these 8 students and no other have been included in the analysis. It will also be good to have a generic idea of how many Chinese students follow the courses of Spanish in the institutions considered here. If, for instance, there is a total of 15 students per year taking classes of Spanish, then a sample of 8 students is quite good in comparison to the whole population. Otherwise, please consider clarifying why it was not possible to include a broader sample size)
2) The part containing the explanation of the different platforms is very clear but a bit too long, so I suggest to reduce it a bit.
3) although there are few or no previous studies on social network use in Chinese learners of Spanish, there are other works addressing this topic with learners of different L1.
A final discussion (at present call Conclusions, section 7) could compare the results presented here with previous studies involving other population. In particular, I suggest to compare the present results with Ahmad, A., Kumar, G., Ranjan, R., & Philominraj, A. (2022). On the role of social media in Spanish language learning in Indian context. International Journal of Emerging Technologies in Learning (iJET), 17(22), 100-115. Indeed, Ahmad et al. considered a similar topic, and they also reach similar results (e.g., the mixed evaluations of social media use for effective learners, or the use of different platforms to reach different goals).
Such a comparison could improve the general engagement of the present article within the community of scholars working on Spanish teaching and learning abroad.
Author Response
Point-by-point response to Comments and Suggestions for Authors
Comments 1: The corpus is indeed quite limited since 8 students couldn't possibly be representative of the whole category of Chinese learners of Spanish. The author explicitely address this poin in section 7.5, but I suggest to already touch this point when presenting the dataset, by also clarifying why these 8 students and no other have been included in the analysis. It will also be good to have a generic idea of how many Chinese students follow the courses of Spanish in the institutions considered here. If, for instance, there is a total of 15 students per year taking classes of Spanish, then a sample of 8 students is quite good in comparison to the whole population. Otherwise, please consider clarifying why it was not possible to include a broader sample size)
Response 1: Thank you for pointing this out. We agree with this comment. Therefore, we have refined section 5.2 of the article. You can find these changes on page 6, lines 276-284.
"At UPV, there are 11 students enrolled in the course this year. However, this Master's program is globally oriented, so not all students are of Chinese nationality. Similarly, at UB, approximately 12 students enrol in the course each year, but only a few are Chinese. Chinese students applying for these master's programs must have at least DELE (Diplomas de Español como Lengua Extranjera) C1 or B2 Spanish proficiency and an interest and ac-ademic background in social network language learning. Therefore, we could only select Chinese students who met these stringent criteria. Although the sample size is limited, these 8 students all meet the rigorous selection criteria, ensuring that their academic backgrounds and research interests are highly relevant to the study. "
Comments 2: The part containing the explanation of the different platforms is very clear but a bit too long, so I suggest to reduce it a bit.
Response 2: Agree. We have, accordingly, condensed sections 2 "Language Learning through Social Media" This section has been reduced by one-third of its original content to enhance clarity and conciseness. The revised content can be found in lines 69-150.
Comments 3: Although there are few or no previous studies on social network use in Chinese learners of Spanish, there are other works addressing this topic with learners of different L1.
A final discussion (at present call Conclusions, section 7) could compare the results presented here with previous studies involving other population. In particular, I suggest to compare the present results with Ahmad, A., Kumar, G., Ranjan, R., & Philominraj, A. (2022). On the role of social media in Spanish language learning in Indian context. International Journal of Emerging Technologies in Learning (iJET), 17(22), 100-115. Indeed, Ahmad et al. considered a similar topic, and they also reach similar results (e.g., the mixed evaluations of social media use for effective learners, or the use of different platforms to reach different goals).
Such a comparison could improve the general engagement of the present article within the community of scholars working on Spanish teaching and learning abroad.
Response 3: Agree. Thank you for your valuable feedback. In response, we have added a comparative analysis in the final part of the Conclusions section (section 7). You can find the specific content on page 16, lines 637-645:
"Although there are currently few studies on the use of social networks for learning Spanish by Chinese students, Ahmad et al. [85] examined the role of social media in Spanish language learning in the Indian context. Their study found mixed evaluations of social media's effectiveness in language learning, with different platforms offering distinct advantages and challenges for achieving various learning goals. Similarly, our study found that Chinese students face challenges but also benefit from the diverse resources and interaction opportunities provided by social media in resource-limited environments. Comparing these studies helps us understand the impact of social media on Spanish language learning across different cultural contexts."
Thank you again for your insightful suggestions.